# Prevention and Control Strategies of African Swine Fever and Progress on Pig Farm Repopulation in China

**DOI:** 10.3390/v13122552

**Published:** 2021-12-20

**Authors:** Yuanjia Liu, Xinheng Zhang, Wenbao Qi, Yaozhi Yang, Zexin Liu, Tongqing An, Xiuhong Wu, Jianxin Chen

**Affiliations:** 1Guangdong Provincial Key Laboratory of Veterinary Pharmaceutics Development and Safety Evaluation, College of Veterinary Medicine, South China Agricultural University, Guangzhou 510642, China; andrewliu@stu.scau.edu.cn (Y.L.); lzx19970420@stu.scau.edu.cn (Z.L.); 2College of Animal Science, South China Agricultural University, Guangzhou 510642, China; xhzhang@scau.edu.cn (X.Z.); xhwu@stu.scau.edu.cn (X.W.); 3Research Center for African Swine Fever Prevention and Control, College of Veterinary Medicine, South China Agricultural University, Guangzhou 510642, China; qiwenbao@scau.edu.cn; 4Heilongjiang Dabeinong Agriculture and Animal Husbandry Food Company Limited, Harbin 150028, China; yangyaozhi@hljdbn.cn; 5State Key Laboratory of Veterinary Biotechnology, Harbin Veterinary Research Institute, Chinese Academy of Agricultural Sciences, Harbin 150069, China; antongqing@caas.cn

**Keywords:** African swine fever, infection, transmission, biosecurity, repopulation

## Abstract

African swine fever (ASF) is a devastating disease in domestic and wild pigs. Since the first outbreak of ASF in August 2018 in China, the disease has spread throughout the country with an unprecedented speed, causing heavy losses to the pig and related industries. As a result, strategies for managing the disease are urgently needed. This paper summarizes the important aspects of three key elements about African swine fever virus (ASFV) transmission, including the sources of infection, transmission routes, and susceptible animals. It overviews the relevant prevention and control strategies, focusing on the research progress of ASFV vaccines, anti-ASFV drugs, ASFV-resistant pigs, efficient disinfection, and pig farm biosecurity. We then reviewed the key technical points concerning pig farm repopulation, which is critical to the pork industry. We hope to not only provide a theoretical basis but also practical strategies for effective dealing with the ASF epidemic and restoration of pig production.

## 1. Introduction

African swine fever (ASF) is a highly contagious viral disease caused by African swine fever virus (ASFV) infection, with morbidity and mortality rates close to 100% [1,2]. This disease was first reported in Kenya in 1921, and several important intercontinental transmissions have occurred since then [3,4,5]. In August 2018, China’s first ASF case was identified in Shenyang City, Liaoning Province [2,6]. After that, ASFV has spread across the country with an extremely rapid rate, leading to a severe reduction in pig population [7], and causing a heavy loss to the pig industry. According to the data released by the Ministry of Agriculture and Rural Affairs of China (MARA) and Office International des Epizooties (OIE), by November 2021, China had reported 203 cases of ASF and culled 1.193 million pigs [8]. At the very beginning of ASFV outbreak in China, more than 50% of the world’s pigs were raised in China; over 99% of the Chinese pig farms were small farms that produce fewer than 500 pigs annually. The biosecurity level in these farms is either very low or next to none [9]. It was estimated that ASF might lead to a loss of 16.5 billion US Dollars in the first year if it broke out in the United States [10]. Importantly, the number of live pigs raised in China is six times that of the United States. Therefore, it can be speculated that the direct economic loss caused by ASF pandemics in China is much greater than that. Furthermore, it has a significant impact on the overall economic development and people’s livelihood. Therefore, there is an urgent need to effectively prevent and control ASF to restore pig production. This review summarizes the key elements in ASFV transmission, prevention, and control and provides key technical insight to pig farm repopulation.

## 2. Three Key Elements in ASFV Transmission

### 2.1. Source of Infection

The source of ASFV is shown in Figure 1. Infected domestic pigs, wild pigs, soft ticks, contaminated feed (including raw materials and swill), water, semen, pork, personnel, vehicles, and tools are the main sources of ASFV [11,12,13,14]. Moreover, stable flies, leeches, and other blood-sucking insects may also be sources of ASFV [15,16]. Data released by MARA reported that between August and November 2018, 34% of ASF outbreaks were caused by hogwash, 46% by people and vehicles, and 19% by live pigs and pork products [11]. It has been reported that among 100 cases of ASFV infection, 42% of the them were caused by swill feeding, 40% by contaminated people and vehicles, 16% by infected pigs and virus carrying products, and 2% by wild boars [17].

Infectious sources carry viruses for a long time, which is one of the difficulties of preventing and controlling ASF. African warthogs, bush pigs, and soft ticks of *Ornithodoros* spp. are the reservoir hosts of ASFV [18,19]. *O. marocanus* carries the virus for 655 days or even 5 years, with other *Ornithodoros* ticks such as *O. porcinus porcinus*, maintaining an ASFV sylvatic cycle between desert warthogs (*Phacochoerus aethiopicus*) and soft ticks [19,20,21]. Furthermore, it remains unclear how long the recovered pigs could carry the virus and how often they could shed the virus. However, it has been reported that recovered pigs can still shed ASFV and infect susceptible pigs 6 months after ASFV infection [22]. ASFV could last for a long period in pig populations due to a constant supply of susceptible pigs [5,23].

The tenacious vitality [19,24] and tough resistance to inactivation of ASFV [5,25,26] are two other aspects that make such epidemics hard to manage. ASFV survives for 11 days in feces at room temperature, one month in contaminated pen, 18 months in blood stored at 4 °C, and several years in frozen meat [27]. Effective disinfection of ASFV can only be achieved when recommended concentration of disinfectant is used and contact time is ensured [28]. It takes 30 min to inactivate ASFV with 0.8% sodium hydroxide, 2.3% chlorine preparation, 3% o-phenyl phenol, 0.3% formalin, and 1% calcium hydroxide [27]. However, the actual duration of disinfection in pig farms is much shorter than that and it is impaired by a variety of organic matters such as protein [28], making it difficult to kill the virus in the environment or on the surface of objects.

### 2.2. Transmission Route

#### 2.2.1. Oral Transmission

Ingesting virus-contaminated feed, drinking contaminated water, and swallowing virus particles from infectious sources are the most important routes of ASFV transmission. In a simulated transoceanic transport study of ASFV from Europe to the United States, living viruses were detected in different feeds, suggesting feed carry infectious virus [29]. Niederwerder et al. demonstrated that pigs that ingested feed contaminated with the Georgia 2007/1 strain were infected with a minimum infectious ASFV dose of 10^4^ TCID_50_ and the median infectious dose was 10^6.8^ TCID_50_ [30]. Surprisingly, the minimum infectious dose of ASFV in drinking water was only 1 TCID_50_ and the median infectious dose was 10 TCID_50_ in the same study [30], indicating that ASFV transmission through drinking water is much more efficient than that via feed. Greig et al. reported that the oral median infectious dose of a highly virulent Tanzania ASFV strain by ingesting contaminated feed was 10^5.4^ HAD_50_ [31]. Up to now, we still know relatively little about factors that are important for ASFV transmission through contaminated feed [13]. Considering that the incubation period of infected pigs is 3–19 days, the virus can be detected in oral fluid on day 2 post infection when a large number of viruses are excreted [2,32]. Therefore, intensive pig farms should put drinking and feeding safety in a particularly important position. The traditional drinking and feeding mode of the connecting trough in pregnant sow unit, which is popular in Asia pig farms, should be changed into a new model of an independent water drinker and feeding bowl.

Several studies reported that viable viruses were detected in nasal fluid, rectal fluid, urine, and other excretions from ASFV-infected pigs [14,26,33]. Montgomery showed in his study that domestic pigs were infected when consuming feces and urine contaminated feed with a virulent Kenya ASFV strain [3]. Moreover, it was reported that 9.7–36.1% of infected pigs showed symptoms of oral, nasal, anal, or vaginal bleeding in the early stage of acute ASFV infection, and the titer of the virus in the blood was very high [2,34]. Therefore, it easily results in environmental contamination, including feed and drinking water, and causes transmission in surrounding pigs. Thus, it is important to identify and cull the infected pigs as early as possible. Furthermore, swill (kitchen leftovers) feeding has been shown as an important way of spreading ASFV in history [35], which is also often an important transmission route during the early ASFV spread in China [11,17]. Moreover, recent epidemiological investigations suggested that fresh grass and seeds contaminated by secretions from infectious wild boars are possible sources of infection for backyard pigs [13]. For instance, eleven ASFV infection cases originating from wild boar have been reported in China, the Russian far east, and northern South Korea [7]. However, the population distribution of Asian wild boar and the epidemiological information of ASFV in the Asian wild boar population remain unclear.

#### 2.2.2. Aerosol Transmission

ASFV-infected pigs shed viruses to the environment through excretions and secretions, and the virus titer in their oral fluid, nasal fluid, feces, and urine is particularly high during the acute phase [36]. When pigs show symptoms of sneezing and coughing, these infectious secretions may be atomized and converted into virus-carrying aerosols. When the feces or urine carrying the virus are dried, the floating dust caused by the movement of animals may also generate virus-carrying aerosols [37]. It was found that the titer of ASFV in the air was positively correlated with the amount of virus excreted from feces. In the acute stage of ASF, when a high titer of ASFV appeared in feces, a high ASFV load in the surrounding air was also detected. However, there was no direct correlation between ASFV load in the air and the viral excretion level in oral and nasal secretions [37,38]. It is suggested that the virus in the air is likely to come from virus-carrying feces.

The half-life of ASFV in the air is 19.2 min (qPCR test, evaluate physical ASFV decay) or 14.1 min (virus titration, evaluate physical and biological ASFV decay) [37]. The virus may persistently float in the air of a pig house or follow airflow to air outlets. There was about 3 log_10_ TCID_50_ equivalent of virus per cubic meter of air. A pig with 25 kg body weight inhaled 15 L of air per minute and was calculated to have exposed to 4 log_10_ TCID_50_ equivalent of the virus every day, which would be enough to cause infection in a susceptible pig [38,39]. Wilkinson et al. demonstrated that ASFV could be transmitted through air, with a maximum transmission distance of 2.3 m between sick and healthy pigs, but the virus was not detected in the air [40]. De Carvalho Ferreira et al. reported that ASFV was detected quantitatively in the air and air outlet of the virus infected pig house [37]. Olesen et al. confirmed that ASFV could be transmitted by aerosols [33]. In conclusion, ASFV can be transmitted in a pig house in the form of aerosols, which might be an important mode of ASFV transmission in pig farms.

#### 2.2.3. Insect-Borne Transmission

ASFV is the only known DNA virus that can be transmitted by vectors [36,41,42]. Thus far, only soft ticks of *Ornithodoros* spp. have been found to facilitate ASFV replication and are the most common vector of the virus [26]. The first documented case of ASFV isolation in ticks (*O. erraticus*) was recorded in Spain in the 1960s [43,44]. Since then, eight *Ornithodoros* species have been found to be involved in the transmission of ASFV [19]. ASFV can be horizontally, sexually, transovarially, and transstadially transmitted in *Ornithodoros* ticks [36,44]. Certain species of *Ornithodoros* ticks can carry these viruses for a long time after infection [44,45]. *Ornithodoros* ticks prefer living in wild boar nests, and adults can live for decades and survive for a long time without eating, which makes *Ornithodoros* soft ticks an ideal ASFV reservoir and maintains ASFV in a sylvatic cycle among desert warthogs (*Phacochoerus aethiopicus*) [19,21]. The *Ornithodoros* ticks that have been confirmed to be able to transmit ASFV to susceptible pigs include *O. coriaceus*, *O. puertoricensis*, *O. turicata*, *O. erraticus*, *O. marocanus*, *O. moubata complex*, *O. moubata porcinus*, and *O. savignyi*. However, their distributions in China have not been reported so far. Other *Ornithodoros* species (*O. tartakovskyi*, *O. tholozani*, *O. capensis Neumann*, *O. papillipes*, and *O. lahorensis Neumann*) were identified in China [45,46], while it is still unknown whether these species participate in the spread of ASFV. It has been reported that *Dermacentor reticulatus*, a kind of hard tick, is also infected by ASFV and could maintain infection for 56 days [19,47]. A new type of ASFV that can infect hard ticks (*D. silvarum* and *D. niveus*) was discovered in China in 2018 and this new type of ASFV could be transmitted across ovaries, from *D. niveus* female adults to the first generation larvae [47]. However, neither of the two studies showed that hard ticks are able to transmit ASFV to susceptible pigs.

Other insects that may spread ASFV have also been reported. For example, the stable flies (*Stomoxys calcitrans*) can be infected by ASFV and maintain a detectable virus number for at least two days [48]. Moreover, studies have confirmed that stable flies can not only mechanically transmit ASFV to susceptible pigs [48], but also transmit the virus through biting. Even ingested ASFV-infected stable flies were reported to cause infections [16]. However, at present, it is not clear what role, if any, the stable flies play in the ASFV epidemics. Furthermore, recent studies demonstrated that ASFV might persist in leeches (*Hirudo medicinalis*) and kissing bugs (Family: Reduviidae, Subfamily: Triatominae) [15,19]. ASFV has also been detected in swine lice (*Haematopinus suis*) collected from experimentally infected domestic pigs [13,49], while fly maggots are not ASFV reservoir hosts and cannot mechanically spread ASFV [50].

#### 2.2.4. Iatrogenic Transmission

ASFV may spread from virus-carrying pigs to susceptible pigs through contaminated medical equipment, such as shared immunization needles, known as iatrogenic transmission [5,27]. Several studies reported that the blood from ASFV-infected pigs carries enough virus to spread infection [2,14,51], so the iatrogenic pathway could play a significant role in ASFV transmission. However, the infection efficiency of this route and its importance in the epidemiology of ASFV remain unclear. It has been observed in China that during the early stage of an ASF outbreak in intensive pig farms, pregnant sows were often affected faster than other pig groups (such as nursery pigs and fattening pigs). This might be related to shared needle contamination during multiple immunizations within the ASF incubation period. Through the measures of abandoning some vaccines, reducing the frequency of mass vaccination, and practicing strictly single needle for vaccinating a single sow, the incidence rate among sows decreased significantly in the recent ASF outbreaks in China.

#### 2.2.5. Semen Transmission

There is no direct evidence to show whether ASFV is transmitted through semen [26]. However, some studies have shown that ASFV can be detected in semen from infected boars [52]. The World Organization for Animal Health (Office International des Epizooties, OIE) promulgated the Terrestrial Animal Health Code, which stipulates that boar semen should not carry ASFV.

#### 2.2.6. Vertical Transmission

Schlafer and Mebus reported that ASFV infection led to abortion in sows under experimental conditions but failed to isolate the virus from the tissues collected from aborted fetuses. It is speculated that the abortion might be caused by infection stress in sows, rather than vertical transmission of the virus itself [53]. Antiabong et al. provided molecular evidence of vertical transmission of the virus. In the study, ASFV DNA was detected in the placenta and fetal organs from sows showing clinical symptoms of ASF, suggesting that vertical transmission of ASFV could occur across placenta [54]. However, so far, no other cases have been reported. We summarize different routes of ASFV transmission as well as their characteristics and transmission efficiency in Table 1.

### 2.3. Susceptible Animals

ASFV mainly infects the members of the pig family (*Suidae*), such as domestic pigs, feral pigs, and wild boars, as well as *Ornithodoros* soft ticks. However, clinical symptoms are only seen in domestic pigs, feral pigs, and European wild boars, while warthogs (*Phacochoerus africanus* and *P. aethiopicus*), bushpigs (*Potamochoerus porcus* and *P. larvatus*), and giant forest hogs (*Hylochoerus meinertzhageni*) are asymptomatic carriers of ASFV and act as reservoir hosts of the virus [3,20,27]. Attempts to artificially infect other animals (cattle, calf, horse, sheep, dog, cat, guinea pig, oxen, hedgehog, hamster, rat, mouse, and various fowls) have failed [3,36,55]. Several studies demonstrated that ASFV could be propagated in rabbits and goats after the agent was modified through multiple experimental infections [55]. Blood samples from rodents and birds collected from ASF-affected farms in Lithuania and Russia were tested negative for ASFV [13]. Chen et al. tested *Dermacentor (Ixodidae)* hard ticks (*D. nuttalli*, *D. silvarum*, and *D. niveus*) and sheep and bovine blood, and detected ASFV DNA segments in *D. silvarum*, *D. niveus*, and sheep blood samples. Further DNA sequence analysis showed it was from a new type of ASFV. Transovarian transmission of this new type ASFV in *Dermacentor (Ixodidae)* hard ticks (*D. niveus*) was confirmed by PCR. The authors believe that this new ASFV strain has a wider range of hosts, such as sheep, bovines, and hard ticks [47]. However, since the researchers did not isolate the virus and conduct animal infection experiments, this conclusion should be treated with caution.

## 3. Prevention and Control Strategies

### 3.1. ASFV Vaccine

Vaccination is one of the best measures to control viral diseases of livestock. However, there is currently no effective ASF vaccine available. ASFV is a large double-stranded DNA virus with a complex structure and large genome (170~190 kb). It encodes a wide variety of proteins (about 170 proteins), including a variety of immune interfering proteins [7]. Although some of the viral proteins are immunogenic, the main antigenic epitopes have not been determined and the exact mechanism of protective response is not clear, which hinders the development of ASF vaccine [7,56]. The development of the ASF vaccine began in the 1960s. Researchers have explored and tested different types of ASF vaccines, including inactivated vaccines [57,58,59], DNA vaccines [60,61], subunit vaccines [62], and viral vector vaccines [63,64]. Unfortunately, almost all of the efforts for developing ASF vaccines have failed. Inactivated vaccines were proven to be ineffective, as they do not seem to induce cellular immunity, even when immune adjuvants were added. The subunit vaccines could not possibly work well when the main neutralizing antigen is yet to be identified. The majority of DNA vaccines only produce a partial or even no protective effect, except one pool of vector vaccines, which has recently shown 100% protection [7,17,65,66]. Interestingly, ASFV live-attenuated virus vaccines (LAVs) will be more promising. For instance, some recent gene-deleted LAVs have shown great potential [8,67,68,69,70]. However, there is no suitable passage cell line to support the production of ASF LAVs. In addition, the differential labelling technique to distinguish ASFV infection from vaccinated animals (DIVA) needs to be developed and the safety concern need to be properly addressed. These are the key factors that currently limit the development of ASF LAVs [7].

### 3.2. Anti-ASFV Drugs

During the past few decades, some compounds, or commercially available drugs with anti-ASFV activity in vitro, have been documented. Colpitts et al. reported that aUY11, an aromatic nucleoside derivative, not only had significant inhibitory activity against viruses such as Influenza A virus (IAV) and hepatitis C virus (HCV) [71], it could also inhibit the proliferation of ASFV in a dose-dependent manner in Vero cells [72]. Furthermore, Freitas and Mottola and others found that fluoroquinolones could inhibit virus replication by blocking DNA-Topo II of ASFV [73,74]. Gallardo et al. discovered that polyphenols such as resveratrol and oxidized resveratrol could inhibit the proliferation of ASFV by suppressing viral DNA replication and late viral protein synthesis [75]. Sánchez et al. reported that amiloride, a drug used in the clinical treatment of edematous diseases, as an effective inhibitor of macrophagocytosis, has significant anti-ASFV activity on Vero cells [76]. However, the research on these compounds only stalled at the cellular level in vitro condition, and their potential effects in ASFV-infected pigs remain to be determined.

### 3.3. ASFV-Resistant Pigs

For a century, scientists around the world have been tirelessly screening and exploring ASFV-resistant pigs, but with little success. The earliest report of ASFV-resistant pigs in the world can be traced back to 1914–1917. Through challenge tests, Montgomery confirmed that ASFV has completely different pathogenicity to domestic pigs and African wild pigs (African bush pigs and warthogs); the challenge of domestic pigs caused extensive heart, lung, spleen, stomach, kidney, and lymphoid tissue damage, resulting in 100% mortality. As for bush pigs and warthogs, there were almost no clinical symptoms and no death cases, except two individuals that died of unknown causes other than ASFV with mild gastroenteritis (one bush pig) and croupous pneumonia of both lungs (one warthog), and there was no other tissue and organ damage [3]. This study firstly confirmed that African bush pigs and warthogs are resistant to ASFV. In fact, African warthog, bush pigs, and giant forest pig, as reservoir hosts of ASFV, can be asymptomatically infected with ASFV [77,78], and show obvious ASFV resistance [79]; however, almost all domestic pigs of all ages and breeds are susceptible to ASFV, causing varying degrees of clinical symptoms [18,80]. Although the difference of ASFV resistance is likely related to the genetic differences of different pig breeds, we still do not know the genetic determinants of ASFV susceptibility [77]. Palgrave et al. compared the genomes of warthogs and domestic pigs and found a difference of the Rel-like domain involved in NF-κb cytokine signal transduction contains protein A (RELA, also known as P65) between these two species, suggesting that this gene difference may be the genetic basis of different susceptibility to ASFV infection between warthog and domestic pig [81]. Lillico et al. used gene editing technology and replaced the domestic pig RELA gene with the warthogs RELA homologous gene [82,83]; however, the results showed that substituting of the warthog NF-κB motifs into the RELA of domestic pigs is not sufficient to confer resistance to ASFV [84]. An early study found that CD163 is a macrophage-specific receptor for ASFV infection [85], suggesting ASFV-resistant pigs might be created by knocking out the CD163 gene. Interestingly, the CD163-deleted pigs were reported to be resistant to PRRSV infection [86]. However, subsequent studies have confirmed that CD163 is not an essential receptor for ASFV infection [87,88], thereby reducing the feasibility of this assumption.

In addition to gene editing technology, another way to breed ASFV-resistant pigs is to collect and screen tolerant pigs that survived in ASFV outbreak countries or regions. Through the verification of scientific experiments, natural ASFV-resistant pigs can be selected. Penrith et al. acquired a group of domestic pigs with higher resistance to ASFV (with high prevalence of circulating antibodies to ASFV) in northern Mozambique to study whether their offspring had heritable ASFV resistance; unfortunately, after the 105 offspring were challenged, 104 of them developed acute ASF and eventually died [89]. In fact, the screening of natural ASFV-resistant pigs is a long-term, large-sample, time-consuming, labor-intensive task, and perhaps also requires a bit of luck.

In March 2020, a research team from China reported the ASFV-resistant domestic pigs LS-2, which was the first time that ASFV-resistant domestic pigs were successfully identified in the world, and this is of great significance for the screening of ASFV-resistant pigs [90]. This study was carried out in a BSL-3 laboratory, through the challenge test of gene type II ASFV virulent strain, combined with the analysis of anti-infective response characteristics, and it was found that LS-2 pigs were significantly resistant to oral infection of 10^6.0^ TCID_50_ ASFV SY18 strain; compared with ordinary domestic pigs, LS-2 pigs showed a significant improvement in survival rate, viremia, clinical symptoms, and antibody response after challenge, and there were also significant differences in the inflammatory factor expression [90].

The report [90] raised another possibility for ASFV prevention and control, i.e., breeding naturally ASFV-resistant domestic pigs. For this reason, it is necessary to further explore the resistance of LS-2 domestic pigs to ASFV: (1) A variety of ASF virus strains, including those from different genotypes, should be used to challenge LS-2 pigs to explore their extensive ASFV resistance. (2) Both the oral route and iatrogenic route are possible ways for pigs to be infected with ASFV; therefore, under the experimental conditions, the virus challenge also needs to use different methods of infection, such as oral administration and intramuscular injection, to comprehensively verify the ASFV resistance of LS-2 pigs under these different circumstances. (3) For the two-way sentinel pig test, two experimental groups were set up: ordinary domestic pigs and LS-2 pigs were raised in the same pen, and one group of ordinary domestic pigs was challenged to observe the impact on the performance of the LS-2 pigs; the other group of LS-2 pigs was also challenged, and the impact on the performance of ordinary domestic pigs observed. (4) The ASFV resistance of the offspring of LS-2 piglets should be evaluated, and to explore whether their virus resistance characteristics can be inherited, it is especially necessary to evaluate whether the hybrid offspring of LS-2 and Duroc have ASFV resistance. (5) The antiviral mechanism of ASFV-resistant pigs needs to be explored at the molecular level, after which a theoretical basis for disease resistance breeding needs to be provided. We summarize the above five research questions into three levels in Figure 2.

### 3.4. Efficient Disinfection

Disinfection is the way to wipe out infectious organisms by using chemical or physical agents [91]. Effective disinfection strategies require full understanding of ASFV, the right disinfectants, disinfection method, working concentration and duration, suitable operating temperature of the disinfectants, and possibly other parameters. Furthermore, a carefully designed pre-disinfection cleaning and a strict post-disinfection monitoring procedure should be considered [92]. Several papers have described the viability of ASFV under various conditions [26,27,28,55,93]. The core information is summarized in Table 2, and different disinfection methods and their common application are shown in Table 3. In general, routine disinfection needs to be performed in areas where the pig farm is in contact with the outside world, such as sales barns, stockyards, staff entrance, gilt pick up area, and so on. Pre-disinfection cleaning is the most crucial element in a disinfection process [91]. On pig farms, when animals cannot be transferred, it is usually ineffective to carry out disinfection. Therefore, the use of an all-in/all-out feeding mode to reduce pathogen circulation in the field and carry out barn disinfection during its “down time” is more practical and effective [92]. Improper operation of foot baths and wheel baths needs to be avoided to achieve an ideal disinfection effect. For example, disinfectants should be refilled every 2–3 days, sheltered from rain, as this will dilute the disinfectant, and situated away from snow to protect from freezing; moreover, manure, mud, or other debris on boots should be completely washed away before soaking them in the disinfectant, and the soaking duration needs to be ensured [92]. In short, successful disinfection needs to carefully consider various aspects, but failed disinfection requires only a small mistake, including the use of over-diluted disinfectant, incomplete cleaning, insufficient contact time, inappropriate temperature, humidity, pH, etc. Finally, the ASFV nucleic acid test must be conducted to monitor the disinfection effect.

### 3.5. High Levels of Biosecurity

Strict inspection and quarantine of pig-derived products must be carried out in customs departments at international airports, shipping terminals, and railway stations to prevent international passengers from bringing in any kind of pork products. Left-over food on international flights, ships, or trains should be properly disposed [36]. Once a farm is confirmed as ASFV-positive, a 3-km protection zone and 10-km surveillance zone should be implemented around the infected farm and the transportation of pigs should be strictly restricted in these areas [98]. The affected pig farm should be depopulated, the culled pigs should be incinerated, deep buried, or composted, and finally, the farm area as well as all equipment should be thoroughly disinfected, cleaned, and dried for at least 40 days [98].

Scientifically designing the structure of pig farms and implementing strict biosecurity measures are prerequisites for effectively cutting off the transmission route of ASFV, which protects susceptible animals from ASFV infection [99,100]. A typical pig farm biosecurity mainly includes eight aspects, as shown in Table 4. Among them, staff entrances and isolation rooms need to be constructed with special consideration. A structural diagram of staff entrance and isolation rooms is shown in Figure 3. The staff entrance is divided into three parts, including a dirty area, transition zone, and clean area, and solid wood benches (barriers) are placed between the two different areas to avoid cross-area contamination. Employees leave their “dirty” clothes, shoes, and hats in the dirty area, wash their hands and take a bath in the transition zone, and then put on clean overalls and boots to move into the isolation rooms where they stay for 2 days. Swabs of the soles, fingers, and hair are collected and tested in a laboratory during this time. If the test results are negative, they will then be granted entry to the farm. As personnel is an important source of ASFV infection [11,17], pig farms must pay attention to the infrastructure construction of the staff entrance and isolation rooms, have strict admission process in place, and ensure its implementation.

## 4. Pig Farm Repopulation

Since the first outbreak of ASF in August 2018, the pig population in China continued to decrease and leading to a severe reduction in the pig population [7]. Farmers have been trying to repopulate pig farms, but most early attempts failed. According to the monitoring data on 400 designated counties across China, issued by MARA, in November 2019, the total number of live pigs across the country rebounded for the first time, an increase of 2.0% from the previous month [102]. More encouragingly, the number of breeding sows achieved an increase of 0.6% month-on-month for the first time in October 2019, and then continued the month-on-month growth for five consecutive months. In February 2020, the number of breeding sows increased by 10.0% compared with September 2019, and by the end of June 2021, the number of breeding sows and the total pig population reached 45.64 million and 439 million, respectively, finally close to a normal year, which reveals that pig farm repopulation in China was successful [103].

Compared with small- and medium-sized pig farms, large pig-raising enterprises have achieved better production recovery due to advantages in capital and technology. For example, the number of sows in Heilongjiang Dabeinong Agriculture and Husbandry Food Co., Ltd. (Heilongjiang, China), one of the largest pig-raising enterprises in China, has increased from 57,000 before the outbreak of ASF to more than 95,000 at present, an increase of 67%. One of the co-authors of this article participated in the repopulation of thirteen pig farms since June 2019. The number of sows in these repopulating farms ranged from 3000 to 5000. By May 2021, all thirteen pig farms have gone into normal production, achieving a 100% repopulation success rate. Pig farm repopulation includes six steps, as shown in Figure 4 and the key technical points are summarized in Table 5. The approach and implementation of pig farm reproduction in China may provide a reference that could be followed by the others worldwide.

## 5. Conclusions

ASFV has a history of more than 100 years worldwide. It is foreseeable that it will continue to threaten the pig industry and related industries in countries around the world for a long time in the future. Although ASF-preventing and -controlling measures are still limited, the experience accumulated in the successful and failed attempts of ASF prevention and control could provide guidance to practitioners. Breakthroughs and progress made in ASF LAVs and vector vaccines, anti-ASFV drugs, and ASFV-resistant pigs, accompanied by the recovery of the pig population in China, will convey positive energy and confidence to practitioners. If we scientifically understand the three key elements in ASFV transmission, ensure pig farms’ biosecurity, and carry out repopulation step by step, the ASFV prevention and control campaign will certainly be successful.

## Figures and Tables

**Figure 1 viruses-13-02552-f001:**
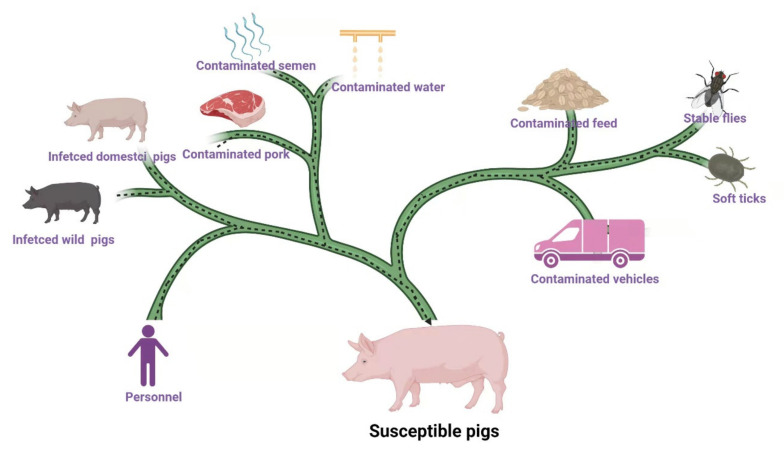
The source of ASFV infection. Infected domestic pigs, wild pigs, soft ticks, contaminated feed, water, semen, pork, personnel, vehicles, and tools are the main sources of ASFV. Stable flies and other insects (leeches, kissing bugs, and swine lice) may spread ASFV. Epidemiological survey showed infected domestic pigs and wild pigs and contaminated feed, personnel, vehicles, and pork are the main sources of infection in China [11,17]. (Copyright agreement number is LR23BRLD53).

**Figure 2 viruses-13-02552-f002:**
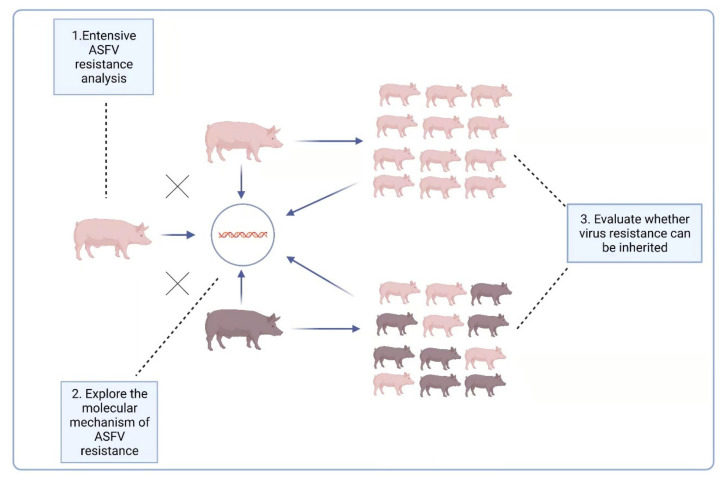
Three levels of LS-2 ASFV-resistance characteristics that need to be further explored.

**Figure 3 viruses-13-02552-f003:**
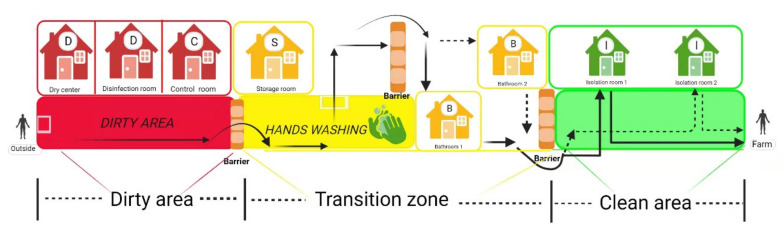
The structural diagram of staff entrance corridor and isolation rooms. Arrows stand for the one-way walking route. Employees leave their “dirty” clothes, shoes, and hats in the dirty area, wash their hands and take a bath in the transition zone, and then put on clean overalls and boots to move into the isolation rooms where they stay for 2 days. Dry center (60 °C, >20 min) and disinfection room (materials that cannot withstand high temperatures, such as special medicines or vaccine products, ozone fumigation, or ultraviolet irradiation [101] can be used to disinfect the materials) are used for goods disinfection. The control room and storage room are used for monitoring the entrance of staff and the storage of disinfected goods, respectively. (Copyright agreement number is EF23BRLLLP).

**Figure 4 viruses-13-02552-f004:**
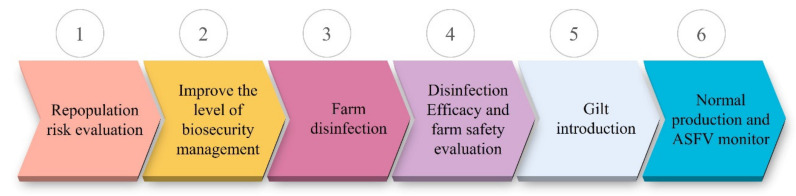
The flow chart for pig farm repopulation. At first, a repopulation risk evaluation should be conducted. Then, relevant facilities are established or renovated to improve the level of biosecurity management. After that, the farm is cleaned and disinfected thoroughly and the disinfection efficacy and farm safety are assessed by laboratory disinfection tests and/or sentinel animal evaluations before gilts introduction. Finally, normal production is carried out and ASFV infections are monitored routinely.

**Table 1 viruses-13-02552-t001:** Different routes of ASFV transmission and their characteristics and transmission efficiency.

Transmission Route	Characteristics	Transmission Efficiency
Oral transmission	Ingesting virus-contaminated feed, drinking contaminated water, or swallowing virus particles.	The most important route of ASFV transmission; transmission efficiency via drinking water is much higher than that via feed.
Aerosol transmission	The titer of ASFV in the air is positively correlated with the amount of virus excreted from feces.	ASFV can be spread in a pig house over a short distance by aerosols.
Insect-borne transmission	ASFV is the only known insect-borne DNA virus; the *Ornithodoros* ticks are the most common vector, though other insects (stable flies, leeches, kissing bugs, and swine lice) may also spread ASFV.	*Ornithodoros* soft ticks are an ideal virus reservoir to maintain the sylvatic cycle of ASFV among desert warthogs and *Ornithodoros* tick species.
Iatrogenic transmission	Virus-carrying pigs and susceptible pigs are immunized or injected with a therapeutic drug with the same needle.	The infection efficiency of iatrogenic transmission and its importance in the epidemiology of ASFV are yet to be fully appreciated.
Semen transmission	ASFV can be isolated from semen of infected boars, but no direct evidence shows that ASFV can be transmitted through semen; the Terrestrial Animal Health Code stipulates that boar semen should not carry ASFV.	Lacking convincing data.
Vertical transmission	Knowledge and data on ASFV vertical transmission are still lacking, except for one study reporting molecular evidence of vertical transmission of the virus.	It is difficult to draw conclusions currently.

**Table 2 viruses-13-02552-t002:** The viability of ASFV under different conditions.

Condition	Viability	Characteristic	Reference
Temperature	37 °C/11–22 days 56 °C/60–70 min 60 °C/15–20 min	Highly resistant to low temperature, but sensitive to high temperature	[26,28]
pH	3.9 < pH < 13.4, with serum/7 days pH 13.4, without serum/21 h pH 13.4, with serum/7 days	Wide range of pH resistance and it can be enhanced by serum	[26,28,94]
Blood	Blood stored at 4 °C/18 months Putrefied blood/15 weeks	Blood enhances the viability of ASFV	[27]
Manure/pen	Feces at 4 °C/8 days Feces at 37 °C/3–4 days Urine at 4 °C/15 days Urine at 21 °C/5 days Urine at 37 °C/2–3 days Contaminated pig pens/1 month	The viability of ASFV in manure is affected by temperature, and low temperature is beneficial to virus survival	[3,95]
Pork/organs	Meat at 4–8 °C/84–155 days Salted meat/182 days Dried meat/300 days Meat with or without bone, ground meat/105 days Cooked meat (minimum of 30 min at 70 °C)/0 days Smoked meat/30 days Frozen meat/1000 days Chilled meat/110 days Offal/105 days Skin/Fat (even dried)/300 days Spleen stored in a refrigerator/>204 days Bone marrow (in boned meat)/180–188 days	Viruses in tissues or organs can survive a long time, and high temperatures are conducive to the elimination of viruses	[26,27]
Feed/Water	Feed, contaminated by infectious blood, 4 °C/30 days Water, contaminated by infectious blood, 4 °C/>60 days contaminated feed, at room temperature/1 day contaminated water, at room temperature/50 days contaminated feed, at 4 °C/>30 days contaminated feed, at 4 °C/>60 days	ASFV survives better in water than in feed	[26,96]
Chemicals/ disinfectants	0.8% sodium hydroxide/30 min 2.3% chlorine (hypochlorites)/30 min 0.3% formalin/30 min 3% orthophenylphenol/30 min 1% calcium hydroxide/30 min	The specified concentration and contact time of the disinfectant is the key to inactivating ASFV	[27,28]

**Table 3 viruses-13-02552-t003:** Different disinfection methods and their common application.

Types	Characteristics	Application
Water	Hot water dissolves inorganic salts, emulsifies fats, washes away organic debris, and easily kills ASFV.	For pig pen cleaning and disinfection, avoid scalding workers or bystanders.
Calcium oxide	Lime wash (calcium oxide mixed with water) has biocidal effects on bacteria and viruses, including ASFV.	Spread on the ground or buried carcasses after depopulation.
Chlorine disinfectants	Concentration, pH, presence of natural proteins, and ammonia affect the efficacy of chlorine-based disinfectants.	Commonly used in water disinfection and sewage treatment in a high concentration, whereas fecal material generally inhibited sodium hypochlorite-based disinfectants.
Iodine and iodine-based disinfectants	Iodophors are combinations of iodine with various carrier compounds. Hard water and organic material reduce the activity of iodophors.	Iodophors are used for general cleaning and disinfection, such as teat dips and surgical scrubs.
Sodium hydroxide	Corrosive and irritating, potential dangers to the environment and to people.	Equipment, vehicle, and sewage disinfection.
Phenolic compounds	Strong odor, enveloped viruses are sensitive to it, as are pigs; small doses could be fatal for pigs.	Use as foot bath disinfectant at the entrances of animal facilities.
Organic acids	Bactericidal and mild viricidal properties make organic acids a good choice of disinfectant in food processing.	For drinking water, feed, and vegetable disinfection.
Formaldehyde	Formaldehyde fumigation can only be completed when the temperature is above 13 °C and the relative humidity is above 70%.	Used for fumigating vehicles, rooms, or even buildings that can be sealed.

The table is adapted from Beltrán-Alcrudo et al., 2017 [27]; Juszkiewicz et al., 2019 [28]; Kahrs, 1995 [92]; Krug et al., 2018 [97].

**Table 4 viruses-13-02552-t004:** Eight concerns of a typical pig farm biosecurity system.

Concern Point	Key Technical Points
Location and layout	The primary principle of location selection for a pig farm is to keep it away from other pig farms, slaughterhouses, residential areas, and transportation lines.
Gilt introduction safety	Pig producers should reduce or stop gilt introduction. Otherwise, ASFV negative gilts must be introduced by air filtration transportation and under strict monitoring.
Set up a fence	A fence around a pig farm can act as a physical barrier to prevent outsiders from entering the pig farm area and to keep animals away from pigs.
Routine disinfection	Effective disinfection requires the right disinfectants, disinfection method, working concentration and duration, suitable operating temperature of the disinfectants, and carefully designed pre-disinfection cleaning and strict post-disinfection monitoring.
Vehicle and goods drying center	ASFV is sensitive to high temperature, and thus, a closed drying room for vehicle and good disinfection at 60 °C (>20 min) is very useful to ensure complete inactivation of ASFV.
Staff entrance corridor and isolation room	Well-designed staff entrances and isolation rooms divided into three parts, including a dirty area, transition zone, and clean area, need to be constructed to reduce the risk of employees bringing in ASFV.
Disposal of sick and dead pigs	Autopsies must be prohibited in or around pig farms and samples of suspected pigs should be collected and tested in a specified facility outside the farm as soon as possible in compliance with the regulations for safe sampling, transportation, and testing of high-risk pathogens.
Feed safety	Stop swill feeding, develop new feed production technology to inactivate possible ASFV existing in feed ingredients or complete feed, and ensure the safety of porcine serum protein powder.

**Table 5 viruses-13-02552-t005:** Key technical points in six steps of pig farm repopulation.

Repopulation Step	Key Technical Points
Repopulation risk evaluation	Analyzing the cause of ASFV outbreak in the farm before and thinking about clearly whether it can be remedied, investigating the ASF epidemic situation around the farm (ASFV re-invasion is often difficult to avoid in farms with defects in location selection).
Improve the level of biosecurity management	Relevant facilities (staff entrance corridor, isolation room, fence, vehicle and material drying center, gilt development unit (GDU), material transfer station, vehicle washing and disinfection center, culled pig transfer room, and feed transfer tower) need to be built or renovated; special biosecurity positions need to be set up, job responsibilities need to be defined, new employees need to be recruited, and regular strict training needs to be carried out.
Farm disinfection	Water disinfection can choose chlorine-containing disinfectants or organic acid, sodium hydroxide can be selected for sewer disinfection, and potassium persulfate can be used for environment disinfection; the disinfection of pig houses can be combined with conventional disinfectants, hot water, flame burning, vacant drying, formaldehyde fumigation. For vehicle disinfection, detergents and disinfectants can be used combined with high-temperature drying.
Disinfection efficacy and farm safety evaluation	Environment and barn cotton swabs are collected and send to the laboratory for ASFV testing to evaluate the disinfection results. Re-stocking with healthy animals should only be undertaken when post disinfection tests and/or sentinel animal evaluations reveal that the premises have a low probability of harboring residual pathogens [92].
Gilts introduction	Gilts come from ASFV antigen and antibody double-negative breeder farms. The use of enclosed and air-conditioned vehicles or vehicles equipped with air filtration systems to ensure the safety of transportation. Gilts should be isolated and observed in the GDU for at least 30 days. Oral fluid and blood samples are collected and tested during this time and ASFV negative results will allow the gilts to be released into the farm.
Normal production and ASFV monitor	Swab samples from all the entry personnel and vehicles are collected for laboratory ASFV testing. Blood and oral fluid samples from diseased pigs and swab samples of ventilation fan blades in pig houses are regularly collected and tested. Once a positive result is detected, it is necessary to activate the corresponding early warning measures and error correction procedures.

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
