# Peer review of "Prevention and Control Strategies of African Swine Fever and Progress on Pig Farm Repopulation in China"

_viruses, 2021, doi:10.3390/v13122552_

Round 1
Reviewer 1 Report
This a good review of on-farm control measures for African Swine Fever. The paper includes some review of the disease and control measures in China, in place since 2018.
There are many very long, complex sentences with several commas. This is confusing to the reader. Also pay attention to preposition choices. Please carefully edit the long sentences, punctuation within and separating long phrases, as well as capitalization.
Line 56 – avoid the words “show” or “showed”. I would suggest reconstructing the sentences without these phrases. It is much more readable and less wordy.
Section 2.3 – please clarify the species of tick and order? Was this the only species examined? There is much speculation in this section and the authors should be very careful of what is stated.
Many of the tables are excellent, but placement within the text should be evaluated so that tables are closer to the relevant sections of the text.
With some editorial revision, this will be an interesting and well-received paper.
Reviewer 2 Report
This review manuscript addresses African Swine Fever Virus, a very important infectious pathogen in the swine industry worldwide, resulting in large reductions in the number of animals with substantial consequences for the meat production. The authors focus on the situation in China, being a major swine producer in the world. The review centers around four main sections: an introduction outlining the major outbreaks in the world forming the rationale for this review; a detailed overview of factors in transmission of the virus and pathways in transmission; a detailed overview on methods to prevent and control transmission; and an overview on the repopulation of the pig industry focusing on the respective process in China.
This review is written in an excellent way, with interesting data for readers involved in the problematics of the virus pandemic for the industry, and readers at some distance from the virus but interested in swine infectious disease in general. The summary tables (sections 2 and 3) are excellent illustrations of what is discussed in the respective text, and by far exceed the issues encountered in coping with the consequences of the ASFV pandemic. Essentially, the overview presented in sections 2 and 3 has a much broader impact, while the content of sections 1 and 4 is more restricted to the virus.
There are a few points that are suggested below to improve the value of this review manuscript, especially to attract readers interested in swine infectious disease in general.
- It is suggested to change the title, in include statements like “African Swine Fever Virus pandemics in the swine population worldwide and its recovery, lessons learned for the swine industry”.
- Then, the abstract should be expanded with statements addressing the impact of the analysis of items addressed in sections 2 and 3.
- Abstract, line 37-38: “Although more than 50% of the world’s pigs are raised in China, over 99% 37 of the Chinese pig farms are small farms that produce fewer than 500 pigs annually”. This was the situation in the past, and has changed during the repopulation (see Chapter 4). It is advised to add this in the abstract.
- In a number of (sub)sections, the authors make conclusions and/or recommendations, which are well phrased. It is advised to bring these at the end of each (sub)section, or alternatively in a specific section at the end, e.g., compile these in section 5.
- Line 299-310: the paragraph on selection of ASFV resistant pigs could be expanded with a statement on economic benefit versus investment. Mentioning “lack of funds” asks for such a statement.
- It could be stressed that the approach and implementation of pig farm repopulation followed in Chana has aspects that could be followed worldwide.
